# Psoriasis and Seasonality: Exploring the Genetic and Epigenetic Interactions

**DOI:** 10.3390/ijms252111670

**Published:** 2024-10-30

**Authors:** Michał Niedźwiedź, Małgorzata Skibińska, Magdalena Ciążyńska, Marcin Noweta, Agnieszka Czerwińska, Janusz Krzyścin, Joanna Narbutt, Aleksandra Lesiak

**Affiliations:** 1Department of Dermatology, Paediatric Dermatology and Oncology, Medical University of Lodz, 90-419 Lodz, Poland; malgorzata.skibinska@umed.lodz.pl (M.S.); magdalena.ciazynska@umed.lodz.pl (M.C.); marcin.noweta@umed.lodz.pl (M.N.); joanna.narbutt@umed.lodz.pl (J.N.); aleksanral.lesiak@umed.lodz.pl (A.L.); 2International Doctoral School, Medical University of Lodz, 90-419 Lodz, Poland; 3Institute of Geophysics, Polish Academy of Sciences, 01-452 Warsaw, Poland; aczerwinska@igf.edu.pl (A.C.); jkrzys@igf.edu.pl (J.K.); 4Laboratory of Autoinflammatory, Genetic and Rare Skin Disorders, Medical University of Lodz, 90-419 Lodz, Poland

**Keywords:** psoriasis, environmental factors, genetics, epigenetics, solar radiation, humidity, air pollution, circadian rhythm, geoepidemiology, seasonality

## Abstract

Psoriasis is a multifactorial, chronic, and inflammatory disease that severely impacts patients’ quality of life. The disease is caused by genetic irregularities affected by epigenetic and environmental factors. Some of these factors may include seasonal changes, such as solar radiation, air pollution, and humidity, and changes in circadian rhythm, especially in the temporal and polar zones. Thus, some psoriasis patients report seasonal variability of symptoms. Through a comprehensive review, we aim to delve deeper into the intricate interplay between seasonality, environmental factors, and the genetic and epigenetic landscape of psoriasis. By elucidating these complex relationships, we strive to provide insights that may inform targeted interventions and personalized management strategies for individuals living with psoriasis.

## 1. Introduction

Psoriasis vulgaris (PsV) is a chronic systemic inflammatory disease that significantly impacts patients’ quality of life [1,2]. Depending on the geographic region, PsV affects from 0.27 to 11.4% of the worldwide population. Ten to forty percent of psoriatic patients are affected with psoriatic arthritis (PsA) [3,4,5,6]. PsV, due to its inflammatory background, is linked with an increased risk of several comorbidities, such as cardiovascular disorders (CVD), metabolic syndrome, nonalcoholic fatty liver disease, and inflammatory bowel disease. Comorbidities in psoriasis may lead to a shortened lifespan for affected patients [7,8,9,10,11]. Due to visible skin lesions and a proinflammatory background, psoriasis is associated with a lower quality of life and psychiatric disorders such as depression and anxiety [12,13,14,15]. Patients affected by PsV experience reduced levels of employment and income [1,12,16]. Beyond geographic differences, psoriasis is affected by environmental factors and presents different seasonal patterns [6,17]. Understanding the seasonal patterns of psoriasis is important for patients and healthcare providers, fostering tailored treatment approaches, optimizing outcomes, and improving patient–clinician relationships. This review aims to elucidate the seasonality of psoriasis vulgaris, exploring potential molecular mechanisms underpinning this phenomenon, and evaluating its implications on treatment strategies and patient adherence.

## 2. Material and Methods

The PubMed, Embase, and Google Scholar databases were searched for the following keywords: “environmental factors”, “genetics”, “epigenetics”, “solar radiation”, “humidity”, “air pollution”, “circadian rhythm”, “geoepidemiology”, “seasonal”, and “seasonality” related to psoriasis vulgaris. Original studies and review papers in the English language, published until 10 March 2024, were taken into consideration. The authors reviewed each paper. Articles that did not mention terms related to the possible seasonal severity of psoriatic symptoms were excluded from further review. The remaining articles’ abstracts were read, and the relevant articles relating to seasonality and environmental factors have been read in full and included in the analysis. The authors also screened the references in the selected articles and included relevant publications. Figures were prepared with BioRender© (www.biorender.com) software.

## 3. Molecular Background of Psoriasis

### 3.1. Genetics of Psoriasis

Psoriasis vulgaris is a multifactorial disorder in which the genetic background has a major impact on the course of the symptoms [18]. About 10 to 40 percent of PsV patients are also affected by psoriatic arthritis [6,17]. PsV and PsA share strong hereditary components. Genes responsible for symptoms may be inherited dominantly or recessively [19]. PsA occurs at a high rate in close relatives, more likely in mono-compared dizygotic twins [20]. Genome-wide association studies (GWAS) revealed more than eighty loci associated with psoriasis, including IL-23/NF-κB/epidermal differentiation signaling [19]. Genes associated with PsA include *HLA-B/C*, *HLA-B*, *IL-12B*, *IL-23R*, *TNP1*, *TRAF3IP3*, and *REL* [19,21]. These genes are associated with the maintenance of cutaneous barrier functioning, control of innate immune responses mediated by the nuclear factor kappa-light-chain enhancer of activated B cells (NF-κB), and interferon signaling, but also adaptive responses linked to the activity of CD_8_ lymphocytes and Th_17_ [19,21]. There are at least fifteen chromosomal regions that contribute to psoriatic disorders, known as Psoriasis Susceptibility (PSORS), numbered from 1 to 15 [22].

The *PSORS1* maps to the major histocompatibility complex (MHC) on chromosome 6p21 and encompasses nine genes, alleles, or haplotypes of human leukocyte antigen (HLA) class I [22,23]. *HLA-C*, Coiled-Coil Alpha-Helical Rod Protein 1 (*CCHCR1*), and corneodesmosin (*CDSN*) genes are highly polymorphic in patients with PsV [24,25]. The HLA-B*27, HLA-B*39, HLA-B*38, and HLA-B*08 are associated with a higher risk of PsA with HLA-B27 and HLA-B*39 earlier onset of PsV and PsA. The HLA phenotypes are also associated with different clinical manifestations of PsA, whereas HLA-B*08 is associated with asymmetric sacroiliitis, peripheral arthritis ankylosis, and increased joint damage, and HLA-B*27 is associated with symmetric sacroiliitis, dactylitis, and enthesitis [26]. The *HLA-C* encodes a receptor that participates in immune responses by presenting antigens to CD_8_^+^ T lymphocytes [27]. The HLA-Cw6 allele can present specific melanocyte auto-antigen (ADAMTS-like protein 5) to CD_8_^+^ T cells [24]. The HLA-C*06:02 is associated with an earlier onset of PsV and a later onset of PsA [28]. Corneodesmosin (CDSN) is involved in the production of the protein responsible for skin desquamation; however, its association with PsV is inconclusive [22,27,29,30]. The *PSORS2* locus contains the *Caspase Recruitment Domain Family Member 14* (*CARD14*) gene, which is highly expressed in keratinocytes [31]. CARD14 is an adaptive protein that takes part in NF-κB signaling, leading to the production of proinflammatory cytokines [31]. Gain of function mutations in the *CARD14* gene are associated not only with psoriasis but also with other inflammatory skin diseases: pityriasis rubra pilaris (PRP) and generalized pustular psoriasis (GPP) [19,32,33].

Other *PSORS4* region maps to chromosome 1q21 include the Epidermal Differentiation Cluster (EDC), which participates in definitive keratinocyte differentiation [22,34]. Deletion of two genes from the *EDC* region, *LCE3B* and *LCE3C*, is associated with psoriasis and disruption of epidermal barrier function [34,35,36].

Polymorphisms associated with PsV are reported in *IL-12B*, *IL-13*, *RUNX1*, *IL-23 receptor* (*il23r*), TNF-regulated protein A20 (*TNFAIP3*), and tyrosine-protein phosphatase non-receptor type 22 (*PTPN22*) genes [37,38,39,40]. Regarding other polymorphisms in *FTO*, *CALCR,* and *AC003006.7* genes were found in patients with PsV that are also related to obesity [41]. Examples of gene polymorphisms associated with psoriasis and their clinical implications are presented in Table 1.

### 3.2. Epigenetics of Psoriasis

For many years, epigenetic factors have been investigated in the context of skin diseases, including psoriasis [47]. Epigenetic modifications influence genetic expression through methylation, histone modifications, and microRNA (miRNA).

#### 3.2.1. DNA Methylation

*OAS2* (2′-5′-oligoadenylate synthetase 2) gene is an antiviral enzyme activated by innate immune system, and its overexpression has been reported in patients with inflammatory diseases. A study by Gu et al. [48] revealed that the *OAS2* gene was hypomethylated in PsV skin samples from 39 psoriatic patients. In another study by Roberson et al. [49], over 1300 gene transcripts were altered in 20 skin samples of the patients with PsV (12 biopsies of skin affected with psoriasis and 8 biopsies of uninvolved skin) compared to 10 samples from healthy volunteers. Out of 1108 different genes’ expressions, 674 genes’ CpG sites were hypermethylated and 444 hypomethylated. The altered CpG site methylation was observed and associated with transcriptional upregulation of genes such as *KYNU*, *OAS2*, *S100A12*, and *SERPINB3,* which are associated with PsV. The authors also reported that after one month of therapy with TNF-α inhibitors in psoriatic patients, the methylation reversed toward the non-psoriatic state observed in healthy subjects. Verma et al. [50] performed genome-wide analysis using epidermal samples to investigate global DNA methylation in patients with psoriasis. Authors mapped over thirty-five thousand differently methylated sites in patients with PsV compared to healthy volunteers. Additionally, a large number of differently methylated sites were present in clinically uninvolved skin samples versus healthy epidermidis, suggesting the presence of a pre-lesion state in clinically healthy skin with no psoriatic symptoms. Chandra et al. [51] profiled the methylation of psoriatic lesions and adjacent normal skin tissues of 24 patients with PsV (48 pairs) and compared to a cohort of 30 skin samples of 15 healthy participants. Genome-wide methylation profiling revealed an inverse correlation between methylation and overlapping gene activity located in *PSOR* regions, including *100A9*, *SELENBP1*, *CARD14*, *KAZN,* and *PTPN*.

#### 3.2.2. Histone Modification

Zhang et al. [52] analyzed histone methylation of psoriatic lesions and peripheral blood mononuclear cells (PBMCs) in thirty patients with PsV and twenty healthy volunteers. The mean total of 5 mC levels of psoriatic PBMC’s were significantly increased compared to normal controls and positively correlated with disease activity assessed with PASI scores. Ovejero-Benito et al. [53] analyzed PBMC samples of 39 PsV patients treated with biologics (ustekinumab, secukinumab, adalimumab, and ixekizumab) before and after the start of the therapy and from forty-two healthy subjects. Psoriasis patients presented reduced levels of acetylated H3 and H4 and increased levels of methylated H3K4 when compared to controls. There were no statistically significant changes in methylation level between pre- and post-treatment samples; however, changes in methylated H3K27 level were found between responders and non-responders to biological medications after 3 months of therapy. Eleven patients in this group also suffered from psoriatic arthritis. Patients without PsA presented significant differences in H3K4 methylation level between responders and non-responders to biological drugs at 3 and 6 months. Those results indicate that histone methylation may be used as a biomarker of treatment response to biological agents among patients with PsV.

#### 3.2.3. Non-Coding RNA

Psoriasis is associated with multiple chromosome loci, and the majority of signals in genome-wide association studies (GWAS) are found in non-coding regions of the human genome [54,55]. Non-coding RNAs (ncRNAs) are RNA molecules that are not translated into proteins but are playing a key role in transcriptional and post-transcriptional DNA activity [54,55,56]. Long non-coding RNAs (lncRNA) are ncRNAs that are longer than two hundred nucleotides, while microRNAs (miRNA) are composed of 18–23 nucleotides. These two major classes of ncRNAs have a significant role in PsV pathogenesis [54,55,56].

MicroRNAs primarily regulate gene expression by binding to the 3′ UTR of mRNA, forming a miRNA–mRNA complex and leading to mRNA degradation [57]. These molecules modulate gene expression by influencing epigenetic modifications. MiRNAs can influence gene silencing and may contribute to human diseases by modulating DNA methylation in CpG islands. Additionally, miRNAs target enzymes that are important for DNA methylation and histone modifications. Patients with psoriasis have significantly higher levels of miRNA expression than healthy individuals, and these molecules may be involved in the pathogenesis of psoriasis. In several studies, miRNA-146a, miRNA-203, miRNA-21, miRNA-31, miRNA-184, miRNA-221, and miRNA-222 were upregulated, whereas miRNA-99a, miRNA-424, and miRNA-125b were downregulated in patients with psoriasis when compared to healthy subjects [51,56,58,59].

LncRNAs act as epigenetic modulators through the recruitment of transcription factors and chromatin-modifying proteins to transcriptionally active loci [60,61,62]. Studies have identified over four thousand lncRNAs that are differently expressed in psoriatic skin compared to non-lesional or healthy skin [8,62].

#### 3.2.4. Seasonality of Epigenetics

While the topic of seasonal epigenetic changes in plants and animals is well explored, the literature on the seasonality of epigenetic factors in human immunology is still limited. Dopico et al. [63] found that 23% of the genome (5136 unique genes out of 22,822 genes tested) show significant seasonal differences in expression. Moreover, they observed an inverted pattern of those expressions when comparing Europeans to Oceanian people. Among others, during winter months in Europe, a proinflammatory profile with high levels of soluble IL-6 receptor and C-reactive protein was observed. The proinflammatory drive of the immune system may be an evolutionary adaptation of humans to difficult environmental conditions during autumn, winter, and early spring. Furthermore, the daily variables of mean ambient temperature and mean sunlight hours both served as linear predictors of seasonality, which suggests human environmental adaptation [63].

### 3.3. Cellular Pathomechanisms in Psoriatic Disease

Psoriasis is considered a T-cell-mediated condition affecting macrophages, dendritic cells, neutrophils, keratinocytes, and other cells, leading to hyperproliferation of keratocytes [64,65,66,67,68]. The process consists of two phases. The first phase is triggered by external factors such as trauma, stress, and infections, which cause the release of deoxyribonucleic acid (DNA) and antimicrobial peptides (AMPs) such as cathelicidin (antimicrobial peptide LL-37), S100, and human β-defensins [67,69,70,71]. These molecules can create a complex binding to Toll-like receptor nine (TLR9) on plasmacytoid dendritic cells in the dermis, causing the release of proinflammatory factors: interferons IFN-α and INF-β, tumor necrosis factor alpha (TNF-α), and interleukins (IL) 6 and 1β. Further reactions lead to involvement and activation of naïve T cells, which secrete TNF-α, IL-12, and IL-23 and cause differentiation into mature T cells. Th_17_, Th_1_, and Th_22_ conduct the further release of proinflammatory cytokines (TNF-α, INF-γ, IL-17, and IL-22) and activation of the JAK/STAT pathway, which further creates a self-generating positive loop causing proliferation of the keratinocytes, acanthosis, and then scaling of the epidermis [68,72,73,74]. It is also hypothesized that because of the polyspecificity of T-cell receptor (TCR), a variety of environmental antigens that align with previously identified potential psoriasis risk factors may interact with a pathogenic psoriatic TCR [75,76]. This interaction could potentially trigger an autoimmune response against melanocytes in psoriasis. Steering clear of these environmental risk factors could aid in the control and management of psoriasis [75,76].

## 4. Environmental Factors Affecting Psoriasis

### 4.1. Sunlight

The Sun is a natural source of electromagnetic energy, from gamma radiation to radio waves. Due to the atmosphere filtering effect, primarily a range between 280 and 1000 nm reaches the surface of the Earth, consisting of 53% of infrared radiation, 43% of visible light, and 4% of ultraviolet radiation [77,78]. Spectra of solar radiation are represented in Table 2.

Approximately 95% of terrestrial UV is UV-A radiation, while the rest is UV-B radiation. The mechanisms of UV-R on the skin are represented in Figure 1.

The significance of sunlight and its impact on psoriasis is well established. Phototherapy is one of the most common treatment options for children and adults [95,96]. Exposure to solar radiation in skin influences several epidermis functions due to apoptosis, DNA damage responses and cell cycle control, innate and acquired immune regulation and inflammation, redox response and angiogenesis, circadian rhythmicity, and keratinocyte differentiation [97]. In keratinocytes of psoriatic skin, UV-B of 311 nm wavelength exposure causes upregulation of pro-apoptotic genes, leading to the apoptosis and suppression of the keratinocyte differentiation. UV-B also causes increased levels of reactive oxygen species (ROS), which can result in oxidative damage to proteins, DNA, and lipids. Excessive production of ROS can disrupt redox homeostasis and cause DNA damage [97]. Due to the above mechanisms, phototherapy (especially UV-B of 311 nm) is one of the treatments for patients with PsV [77,95,97,98]. Photochemotherapy with psolaren and UV-A irradiation (PUVA) used both locally (e.g., PUVA baths) and orally is another standard phototherapy used in psoriasis treatment [77,95]. Lately, the spectra of visible light, the blue and red light, were used in psoriasis treatment; however, the results of the clinical studies are inconclusive [99,100,101,102,103,104].

Heliotherapy is defined as medical therapy involving exposure to natural sunlight [105]. It was reported that heliotherapy may be an alternative to other treatments for psoriasis in low- and mid-latitude regions of the Earth, including cold and cloudy countries such as Poland [78,106,107]. In Italy [108] and Israel [109,110], heliotherapy has been used in practice. However, UV radiation, despite its therapeutic effect in patients with psoriasis, has many negative effects on the skin. Exposure to ultraviolet radiation is a major risk factor for the development of solar lentigines and, more significantly, skin neoplasms [81,111,112,113]. The practical application of heliotherapy can be realized using UV intensity at ground level from weather forecasting models [107,114] and/or measurements with standard UV monitoring instruments placed at the tanning site [115].

However, exposure to UV is not beneficial for all PsV patients. Photosensitive psoriasis (P-PsV) refers to a group of patients who may experience a worsening of their psoriasis symptoms when exposed to sunlight [116]. This phenomenon affects between 5.5% and 24% of psoriatic patients with female predominance and low age of disease onset [116,117]. The main cause of this phenomenon is Koebnerization, which is the development of new psoriasis lesions due to local trauma, such as UV radiation and sunburns [116,118]. Other factors that can exacerbate symptoms include coexisting conditions like polymorphic light eruption, systemic erythematous lupus, porphyria, and chronic actinic dermatitis. *HLA-CW*0602* and *CARD14* mutations have been linked to P-PsV [116,117,119]. The innate immune response, triggered by UV-induced damage-related molecular patterns (DAMP), impaired self-coding RNA, or possible inflammasome formation and activation, creates a skin microenvironment that is conducive to psoriasis [116,120]. This environment is characterized by an abundance of IL-17, IFN-γ, and TNF-α and a deficiency of IL-4 and IL-10, leading to worsening psoriatic symptoms [116,121].

### 4.2. Humidity

Psoriasis is associated with increased transepidermal water loss (TEWL) and decreased subcutaneous water content [122]. The reduced water content in epidermis was reported to be related to the papulosquamous skin features of PsV [122,123,124]. In a study conducted by Nakahigashi et al. [123], comparing *aquaporin 3* (*AQP3*) expression via immunofluorescence and skin hydration in the epidermis (stratum corneum hydration—SCH) in 19 PsV patients and ten healthy volunteers was investigated. Results of this study indicated that patients with PsV, when compared to healthy subjects, had significantly increased TEWL and decreased SCH. Voss et al. [125] evaluated ten healthy and 10 PsV skin samples by immunohistochemistry using antibodies recognizing AQP3 and enzyme phospholipase D2, which also interact functionally in normal skin by inhibiting keratinocytes’ hyperproliferation. In psoriasis samples, APQ3 was mainly observed in the cytoplasm rather than the cellular membrane, and PLD2 staining revealed decreased immunoreactivity and aberrant localization. Another cross-sectional study by Montero-Vilchez et al. [124], among others, included 157 healthy individuals and ninety-two psoriasis and compared TEWL, SCH, and temperature values between healthy skin and psoriatic skin. The results of this study revealed statistically decreased values of TEWL and temperature, with increased values for SCH in patients with PsV compared to healthy volunteers. Thus, the authors stated that TEWL and skin temperature measurement can be helpful in the assessment of disease severity and treatment intensity. Similar conclusions were presented in the research paper by Nikam et al. [126]. Denda et al. [127] on mice models (HR-1) demonstrated the influence of prolonged dry and moist environments on the skin. Exposure to low humidity for 48 h led to increased inflammatory markers and skin barrier disruption, resulting in marked epidermal hyperplasia. Cravello and Ferri [128] measured TEWL changes in the skin of six young female participants who were affected by different environmental factors in the climate chamber. The authors demonstrated the correlation between TEWL and ambient temperature, while the relative humidity had a weaker effect on TEWL in the temperature range under investigation. Mean skin temperature showed a higher correlation with ambient temperature compared with relative humidity.

On the other hand, according to Liang et al. [129], the incidence of biologic therapy initiation in patients with psoriasis appeared to be higher in low-humidity regions of the South and Midwest USA than in other regions.

### 4.3. Air Pollution

Particulate matter (PM_2.5_: ≤2.5 μm; PM_10_: ≤10 μm) and nitrogen dioxide (NO_2_) cause oxidative damage to the epithelial cells by production of volatile organic compounds, increasing TEWL [113,130,131,132]. According to the available data, PM_2.5_, PM_10_, and surface ozone (O_3_) may play essential roles in psoriasis development via aryl hydrocarbon receptor (AHR) activation, leading to increased Th-17 differentiation [133,134,135,136]. AHR is a ligand-dependent transcriptional factor that plays a crucial role in Th_17_ functioning, influencing the production of IL-22 and taking part in chemical sensing of the circadian rhythm and the skin’s adaptive response to environmental stimuli by controlling the T_reg_ and Th_17_ cell differentiation [113,131,132,134,137]. This ligand regulates the T_reg_ and Th_17_ cell differentiation, playing a vital role in the development of psoriasis. The AHR is modulated by the novel drug, tapinarof, which was recently approved by the FDA in the topical treatment for psoriasis [138]. Other outdoor pollutants such as carbon monoxide (CO) and sulfur dioxide (SO_2_) also cause epidermal damage due to ROSs [113,139]. Air pollutants present seasonal distribution [140,141]. It was shown that increased concentrations of air pollutants cause flares of PsV [142,143]. Liaw et al. [143] reported that the blood concentration of cadmium was statistically significantly higher in patients with psoriasis and correlated with the disease severity.

Air pollution may influence psoriasis flare-ups due to its seasonal pattern caused by indoor heating and decrease beneficial UV exposure for psoriatic patients [113,144].

Another air pollutant, cigarette smoke, is strongly associated with the incidence and severity of psoriasis [133,145]. Nicotine causes an increased secretion of proinflammatory cytokines IL-2, IL-12, IFN-γ, and granulocyte–monocyte colony-stimulating factor, which also participate in the keratinocyte’s differentiation [146]. Shan et al. [147] showed that CD1a^+^ antigen-presenting cells (APCs) from the lungs of patients with emphysema could induce autoreactive, pathological T helper 1 (Th_1_) and Th_17_ cell responses and cause overexpression for IFN-γ and IL-17A, which play a crucial role in psoriasis pathogenesis. Smoking is also related to two single nucleotide polymorphisms at the *CSMD1* gene (rs7007032 and rs10088247), which effect epithelial cell turnover and influence the differentiation of keratinocytes [39].

PsV patients have an increased risk of pulmonary diseases such as asthma, chronic obstructive pulmonary disease (COPD), obstructive sleep apnea (OSA), and pulmonary hypertension [148]. Similarly, pulmonary disorders, such as COPD [149,150], asthma [151], and OSA [152], are related to the higher risk of psoriasis susceptibility. Smoking is also strongly associated with psoriasis [153]. Zhou et al. [154] conducted a meta-analysis of the relationship between smoking and treatment efficacy in PsV patients. The authors concluded that smoking negatively affects the efficacy of psoriasis treatment and smokers are exposed to more immune regulators compared to non-smokers. These results indicate that smoking causes worse improvement than conventional and biological treatments. On the other hand, smoking is associated with an increased incidence of psoriasiform lesions in inflammatory bowel disease patients undergoing anti-TNF therapy [155].

### 4.4. Circadian Rhythm

Li et al. [156] reported a higher risk of incidence of psoriasis and psoriatic comorbidities in rotating night shift healthcare workers. This phenomenon can be explained by lower exposure to sunlight, decreased vitamin D levels, reduced production of melatonin, or a tendency to behavioral disruptions, but also by the circadian rhythm’s disturbance. A similar study by Huang et al. [157] had comparable results in urticaria incidence.

The circadian rhythm is regulated by the so-called master clock located in the suprachiasmatic nucleus (SCN) and affects molecular clocks in peripheral tissues [158,159,160]. The molecular mechanism of the circadian rhythms is based on the transcription–translation positive and negative feedback loops of circadian clock genes and their proteins [161]. The positive arm of the cire clock gene network consists of the transcription factor Brain and Muscle ANT-like 1 (*BMAL1*) and Circadian Locomotor output Circle (*CLOCK*), which form heterodimers binding the enhancer box to active transcription of Period (*Per*) 1, 2, and 3 and Cryptochrome (*CRY*) 1 and 2. PER and CRY drive the negative arm by inhibiting the expression of *BMAL1-CLOCK* genes, leading to the impediment of their transcription. Decreased levels of PER and CRY factors conduct the BMAL1-CLOCK activation, completing the circle within 24 h. Other clock-controlled genes associated with circadian rhythms within the SCN are RAR-related orphan receptors (RORs) and REV-ERBS-response elements (RORE), which form a secondary feedback loop influencing the oscillatory expression of the *BMAL1* gene [160].

The light causes the retina’s neuronal and hormonal signaling, activating the SCN [158]. Neurotransmitters involved in this process, vasoactive intestinal peptide (VIP) and arginine vasopressin (AVP), along with the astrocytes, stabilize the circadian clocks within the SNC [158,159,160]. Peripheral tissues’ clocks, for example, skin, are also influenced by environmental factors, such as temperature, activity, and food intake [162,163].

The increased risk of psoriasis appears to be linked to disturbances in circadian rhythms [156,160,164,165,166,167,168]. The direct link between Th_17_ cell differentiation and the circadian clock was first described by Yu et al. in 2013 [169]. Expression of *NFIL3* suppresses Th_17_ cell development by repressing orphan receptor RORγt transcription. *CLOCK*, *PER2*, and *BMAL1* genes also play a significant part in the transcription of IL-23R in γ/β^+^ T cells. Mutations in these regions are associated with the induction of psoriatic symptoms [165]. BMAL1, CRY, RORα, and REV-ERBα proteins are positive regulators of the anti-inflammatory reactions [165]. Downregulation of these proteins enhances inflammation. CLOCK protein, however, can promote inflammation via the NF-κB pathway. Circadian genes such as *CRY2*, *PER3*, *NR1D1*, and *RORC* are downregulated in psoriatic lesions and the adjacent normal skin compared to the skin from non-psoriatic individuals [170]. The mRNA expressions for IL-17A, IL-22, and IL-23 are strongly associated with circadian genes and elevate at night, decrease at dawn, and then increase during the day [165]. The disruption of the *REV-ERBα* gene, which suppresses RORγt-driven Th_17_ cell differentiation, also seems to be involved in developing psoriasis and psoriatic arthritis symptoms [171].

A study conducted by Hirotsu et al. [172] on the mice Balb/C went through selective paradoxical sleep deprivation, which led to immunological disturbances in mice. The authors observed increased levels of proinflammatory cytokines IL-1, IL-6, and IL-12 and decreased levels of the anti-inflammatory cytokine IL-10 in tested mice. Another study investigated the potential reciprocal relationship between the circadian clock, feeding time, and skin inflammation exacerbation [173]. Mice lacking circadian rhythms had more significant epidermal hyperplasia and more robust activation of the IFN pathway. Also, the daytime-restricted feeding shifted the phase of IFN-sensitive gene expression in mouse skin.

Németh et al. compared the expression of core clock genes in six human lesional and six non-lesional skin samples and in human low calcium temperature (HaCaT) keratinocytes after stimulation with proinflammatory cytokines [165]. The authors also assessed the CLOCK proteins in skin biopsies from PsV patients by immunohistochemistry. Altered *CLOCK* gene expression was observed in non-lesional psoriatic skin, with increased *CRY1*, *BMAL1*, *PER1,* and *PER2* gene expression and decreased *REV-ERBα* expression. Moreover, cytokine treatment affected circadian oscillation and relative mRNA expression of the clock gene in HaCaT keratinocytes. In lesional psoriatic skin, *REV-ERBα* and *CRY1* genes showed altered rhythmicity and reduced relative mRNA expression compared to healthy skin.

Melatonin (MLT) has a crucial impact on the circadian clock and is also related to psoriasis etiopathogenesis [174,175]. Some studies showed that patients with psoriasis, compared to non-PsV volunteers, presented lower MTL levels [176,177,178].

Nguyen et al. [179] performed a prospective randomized study comparing the efficacy of topical corticosteroids in forty-six psoriasis patients based on the time of their application; one group applied medications between 5 p.m. and 6 p.m., and the second group between 8 a.m. and 9 a.m. In the group of patients who applied for the treatment in the evening, the response to the therapy was faster compared to the patients who applied the medication in the morning. However, these differences were statistically irrelevant after five days of treatment.

## 5. Geoepidemiology of Psoriasis

The seasons are different depending on the latitude. In the latitudes of tropical and subtropical climate zones, the weather is usually less dynamic and variable than in temperate zones. Those differences connected to exposure to different humidity, temperature, sunlight, etc. may also influence the severity and seasonality of psoriasis. Reports on the geoepidemiological profile of PsV are limited, with only 19% of the countries having epidemiological data on the disease [180]. Most studies were conducted in Europe, North America, and Australasia. In 2020, The Global Burden of Disease (GBD) analyzed the prevalence and impact of skin conditions [181]. Among the fifteen most common skin diseases, psoriasis was assigned the sequelae of itch and disfigurement. The authors also emphasized the concern about the pathogenic association between PsV and cardiovascular disorders. A study by GBD showed an increased number of psoriasis diagnoses between 1990 and 2017. In addition, the risk of psoriasis assessed in the above study increased with age, which is consistent with literature data. Parisi et al. [180] conducted a meta-analysis and systemic review of the worldwide epidemiology of psoriasis based on 159 studies. The studies differed in the prevalence of PsV regionally and nationally. Regionally, the occurrence of the disease in the overall population varied from 0.11% in East Asia to 1.58% (from 0.50% to 5.73%) in Australasia and 1.52% (from 0.87% to 2.74%) in Western Europe. Country-specific prevalence of psoriasis varied substantially. Considering the estimate for the overall population, Australia 1.88% (from 0.59% to 6.10%), Norway 1.86% (from 0.94% to 3.97%), Israel 1.81% (from 0.83% to 4.44%), and Denmark 1.79% (from 0.91% to 3.61%) had the highest estimates of the prevalence of PsV. The estimated prevalence of psoriasis in countries in East Asia is significantly lower. Taiwan, with a PsV prevalence of 0.05% (0.02% to 0.16%), seems to be the country with the lowest number of cases of psoriasis per 100,000 people worldwide. The prevalence was highest in high-income countries, which may also have influenced the study results due to better healthcare systems, disease awareness, and better data quality. Another interesting study conducted by Lecaros et al. [182] described that the incidence of psoriasis in Chile depended on the latitude. The data revealed a gradual elevation of prevalence in the population living in the country’s southern parts when compared to northern areas. Among many factors that could have impacted this phenomenon, the authors mention environmental ones.

## 6. Seasonality of Psoriasis

Clinically observed seasonality of medical conditions is a well-known phenomenon [183]. The relationship between environmental factors presenting seasonal patterns, genetic background, epigenetic modifications, and inflammatory reactions may influence the activity of psoriasis symptoms (Figure 2). The differences in the activity scores of the diseases seen among different countries during the year may not solely be influenced by environmental factors but also the cultural activities and festivities. For example, intermittent circadian fasting (Ramadan), a common Muslim practice, seemed beneficial in patients with psoriasis and psoriatic arthritis [184,185].

There is a common belief that in most patients with psoriasis, skin lesions improve in warmer months of the year [183,186,187,188,189,190,191,192] (Figure 3). However, not all literature data support this statement (see Table 3 for more details) [189,193,194,195].

**Figure 2 ijms-25-11670-f002:**
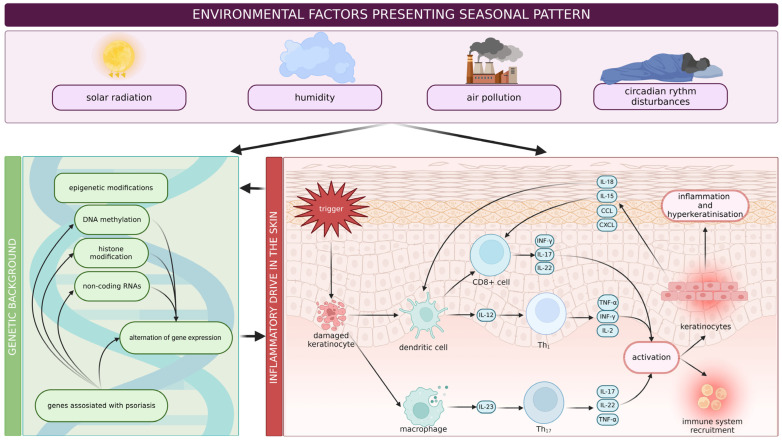
Relation between environmental factors, genetic background, epigenetic modifications, and inflammatory reactions leading to deterioration of psoriatic skin symptoms. Environmental factors presenting seasonal patterns, such as solar radiation, humidity, air pollution, and circadian rhythm disturbances, influence genetic background and inflammatory drive within the skin [77,128,133,139,157,160,167]. Genes associated with psoriasis may be altered by environmental factors through epigenetic modifications [54,56,58,162,196]. Inflammation also causes a shift in gene expression, which leads to a positive feedback loop [66]. Inflammatory reactions cause hyperkeratinization and skin symptoms [66]. This figure was created with BioRender.com.

**Figure 3 ijms-25-11670-f003:**
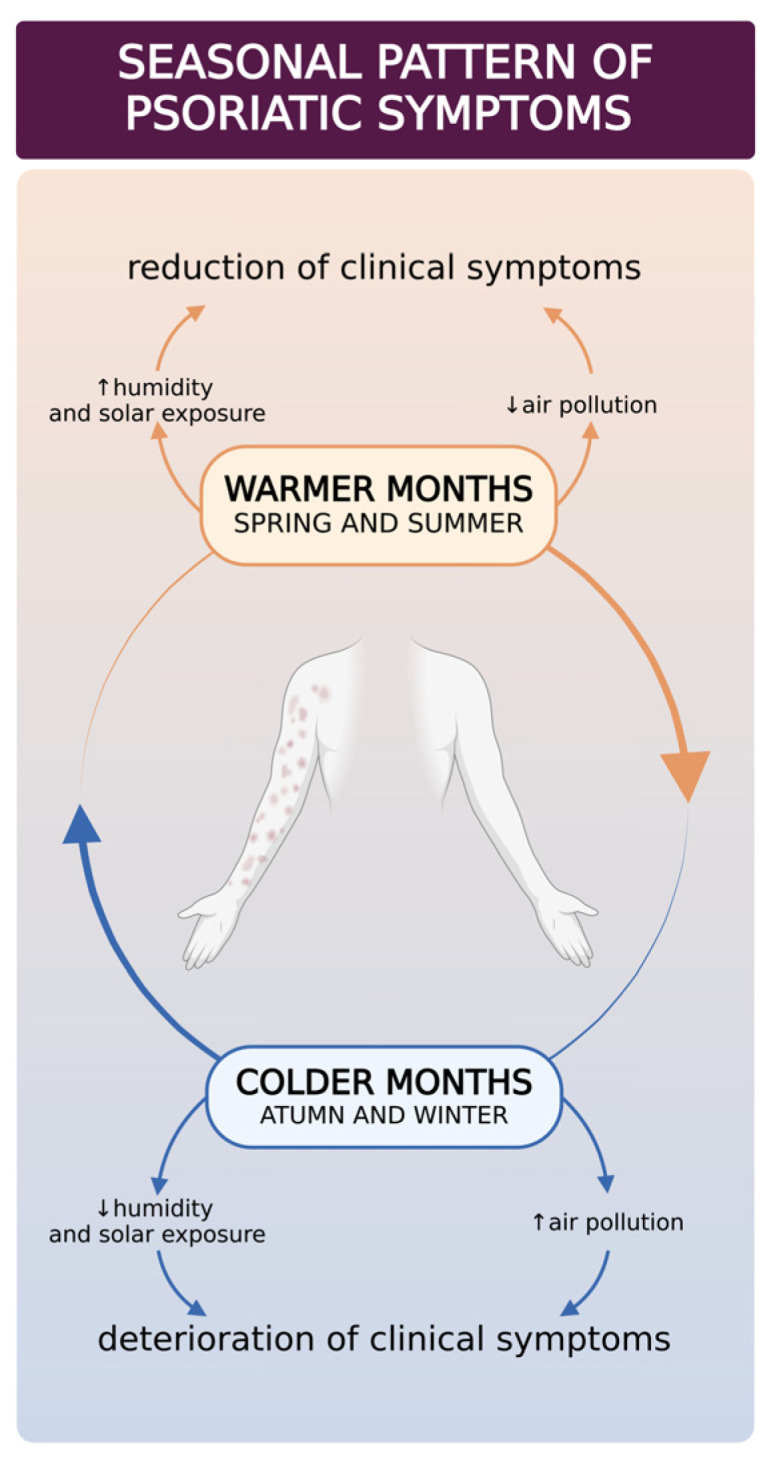
Factors leading to seasonal patterns of clinical deteriorations and improvements in patients with psoriasis. Due to the changes in environmental factors, such as humidity, solar exposure, and air pollution, patients may experience a reduction or deterioration of clinical symptoms. This figure was created with BioRender.com.

Brito et al. [187] examined the seasonality of the hospitalizations at a dermatological ward, with no differences observed regarding patients with psoriasis. A study by Ferguson et al. [188] revealed that 77% of patients with psoriasis reported seasonal changes in the disease activity with exacerbations in winter (67%) and summer months (24%). Mrowietz et al. [197] conducted a questionnaire study in northern Germany. Five hundred thirty-six patients were self-classifying into four types of psoriasis intensity during a year: type 1—stable disease course with no variation depending on the season (40.86%); type 2—unstable disease course with no variation depending on the season (22.57%); type 3—increased flare incident rate in winter (30.6%); and type 4—increased flare incident rate in summer (5.97%). In a retrospective study of 20,270 Chinese patients with psoriasis, the disease duration, hyperlipidemia, and smoking were associated with severe psoriasis in autumn and winter [198]. In this study, age and occupations with extensive sunlight exposure were negatively associated with seasonal psoriasis exacerbation. Considering the findings of Jensen et al. [199], only 30% of psoriatic patients in northern and central Europe reported improvement in their disease activity in the summer months; however, other factors could be responsible for the results obtained in this study.

A survey study conducted on 1080 Polish patients with PsV revealed that seasonal changes were reported to have a considerable impact on the psoriasis disease course in 45.09% of patients [153].

Internet searches connected with psoriasis and its treatment also showed statistically significant seasonality peak in late winter [195]. On the other hand, Kubota et al. [194] did not observe any difference in seasonal frequency in the number of patients with psoriasis using health services in Japan.

Outdoor conditions for the mid-autumn/winter and mid-spring/summer seasons are very different in Europe, with higher temperatures and stronger solar radiation in the latter period [200]. In mid-spring/summer, it is evident that the patients have a chance to synthesize large amounts of vitamin D during outdoor activities due to UV-B exposure. They also receive more solar radiation in UV-A (315–400 nm) and visible spectrum (400–700 nm), and prolonged outdoor exposures until late evening could interfere with melatonin production.

However, studies do not support the positive effect of possible higher levels of vitamin D on psoriatic patients in the summer months. Theodoridis et al. [201] performed a meta-analysis of the influence of vitamin D supplementation on the severity of psoriasis, which could not confirm its beneficial effect. The Medical Board of the National Psoriasis Foundation in the USA does not recommend oral vitamin D in psoriatic patients with normal vitamin D levels [202].

Considering other environmental factors, it was reported that humidity significantly influenced the epidermis structure that can contribute to seasonal deteriorations and improvements of inflammatory dermatoses such as atopic dermatitis and psoriasis [127].

Another possible explanation for the phenomenon is the seasonally fluctuating expressions of several genes. Ruano et al. [203] analyzed a group of patients with moderate to severe plaque psoriasis treated with etanercept or adalimumab who, after achieving a significant response to the treatment, had a temporary suspension of the treatment. The authors found that the risk of relapse and duration of the remission were related to the time of the year the treatment was stopped.

The seasonality of psoriasis activity may also affect the time when the systemic treatment is started. Liang et al. [129] evaluated initiation, discontinuation, and switching biologics and systemic non-biologic drugs in 74,960 patients with psoriasis between 2016 and 2019 in the United States. The initiation of the treatment peaked in the spring months, followed by the summer, fall, and winter. Discontinuation of biologic drugs peaked in summer, and switching of biologics was highest in spring. The authors underlined that the initiation, discontinuation, and switching of the biologic treatment for psoriasis was associated with seasonality patterns, although seasonality patterns are less clear for nonbiologic systemic medications. The question remains whether the environmental factors may influence the treatment course. In a preliminary study by Niedźwiedź et al. [204], sixty-two patients with moderate to severe psoriasis were evaluated at the beginning of the biological treatment, after 1, 4, and 7 months of therapy. This study involved categorizing patients into two distinct groups: those who initiated treatment during the colder months (from November to March) and those who began therapy in the warmer period (from May to September). After 1 and 4 months of treatment, it was observed that patients on IL-12/23 and IL-17 inhibitors had better improvement when the therapy was started in the summer months. Interestingly, the course of psoriasis improvement remained consistent for patients using TNF-α inhibitors, irrespective of the season. The treatment outcomes after 7 months of therapy were comparable between both seasonal groups and across various types of biologics used.

The research papers related to the seasonality of psoriasis are summarized in Table 3.

**Table 3 ijms-25-11670-t003:** Research on seasonality of psoriasis.

Author(s)	Year of Publication	Study Design	Region	Number of PsV Patients	Conclusion
Lane and Crawford [205]	1937	retrospective analysis of clinic visits	USA	231	Seasonal pattern in 75% of patients. Deterioration of psoriasis in 14.3% of patients in summer, improvement in 60.2% in summer.
Lomholt [206]	1954	personal interview by the investigator	Faroe Islands	206	Seasonal pattern of psoriasis in about 50% of cases. Patients with seasonal patterns observed deterioration in winter and spring (25% and 52%, respectively) and improvement in summer (63%).
Hellgren [190]	1964	analysis of inpatients with psoriasis	Sweden	255	Seasonal pattern of psoriasis in about 50% of patients. Patients observed improvement in winter and summer (7.9% and 22.6%, respectively).
Bedi [186]	1977	analysis of outpatients with psoriasis	northern India	162	No seasonal pattern in 54% of patients. Deterioration in winter and improvement in summer in 25%. Improvement in winter and deterioration in summer by 12%.
Könönen et al. [207]	1986	survey/questionnaires	Finland	1517	Deterioration of psoriasis in 54% of patients in winter, 18% in spring, and 2% in summer.
Knopf et al. [208]	1989	survey/questionnaires	Germany	390	Deterioration of psoriasis in 37.4% of patients in winter, 42.3% in spring; improvement in 60.5% of patients in summer.
Park and Youn [191]	1998	survey/questionnaires	Republic of Korea	870	Deterioration in winter was reported by 65% of patients, and no seasonality or improvement was reported by 35%.
Hancox [189]	2004	retrospective analysis of office visits	USA	no data	No seasonality is observed by using an astronomical calendar. Significant differences were observed by using the meteorological calendar with the majority of visits in spring.
Kubota et al. [194]	2015	statistical analysis of the data in the Japanese national database of health insurance claims	Japan	429,679	No seasonal pattern was observed.
Pascoe and Kimball [192]	2015	analysis of dermatologists’ billing sheets based on PGA scores	USA	5468	The percentage of patients with clear/almost clear disease was highest in summer at 20.4% and the lowest in winter at 15.3%. The number of patients with moderate/severe psoriasis was highest in winter at 40.5% and the lowest in summer at 34.1%.
Harvell and Selig [193]	2016	retrospective analysis of dermatopathological data	USA	223	No seasonal pattern in histopathological diagnosis was observed.
Brito [187]	2018	retrospective analysis of ward admissions	Brazil	155	Percentage of admissions of PsV patients: 29% in autumn, 27% in winter, 25% in spring, and 19% in summer.
Kardeş [209]	2019	analysis of Google Trends queries for psoriasis	USA, UK, Canada, Ireland, Australia, and New Zealand	no data	Statistically significant seasonal pattern of searches for psoriasis with peaks in winter/early spring and troughs in summer/early fall. Peaks in late winter/early spring and troughs in late summer/early fall presented approximately with a 6-month difference between hemispheres.
Wu [195]	2020	analysis of Google Trends queries for psoriasis	Australia, New Zealand, USA, Canada, UK, and Ireland	no data	Significant seasonal pattern for psoriasis, with peaks in late winter/early spring and lows in late summer/early autumn.
Ferguson et al. [188]	2020	cross-sectional online survey	world-wide	186	Seasonal pattern of PsV exacerbation reported by 77% of responders. Deterioration of psoriasis was reported by 67.1% of patients in winter, 23.8% in summer, 7% in spring, and 2.1% in autumn.
Jensen [199]	2021	systematic review	northern and central Europe	12,900	Thirteen publications: nine published before 1958, including the four publications reported above (Lomholt [206], Hellgren [190], Könönen et al. [207], and Knopf et al. [208]). No seasonality in 50% of patients. Approximately 30% improved in summer, and 20% performed better in winter.
Purzycka-Bohdan et al. [153]	2022	national survey study	Poland	1080	Seasonal changes were reported by 45.09% to have a considerable impact on the psoriasis disease course.
Liang et al. [129]	2023	retrospective ecological study of individuals with psoriasis identified in the Optum Clinformatics Data Mart	USA	74,960	The initiation of the treatment peaked in the spring months, followed by the summer, fall, and winter. Discontinuation of biologic drugs peaked in summer, and switching of biologics was highest in spring. Season was associated with initiation, discontinuation, and switching, although the seasonality pattern is less clear for nonbiologic systemic drugs.
Niedźwiedź et al. [204]	2024	retrospective analysis of patients treated with biologics depends on the starting point of the therapy	Poland	62	Seasonality appeared in the effectiveness of IL12/23 and IL17 inhibitor therapy in moderate to severe psoriasis, with better results obtained within the first months of treatment in patients starting therapy in the warm period of the year (May–September). No seasonality was observed in patients treated with TNF-α inhibitors.

Abbreviations: PsV—psoriasis; IL—interleukin; TNF-α—tumor necrosis factor α.

## 7. Conclusions

Psoriasis activity and severity are affected by epigenetic and environmental factors such as sun exposure, humidity, air pollution, and circadian rhythm. The seasonality of psoriasis may be related to a different expression of genes with a more proinflammatory immune system and various environmental changes observed as the evolutionary adaptation of humans to more difficult environmental conditions during autumn, winter, and early spring in temperate and subpolar climate regions. Patients who are exposed to a lower UV dose, a higher amount of air pollutants, or lower humidity may be more prone to deterioration of skin lesions. Also, disturbances in the circadian rhythm, such as changes in sleep pattern, variable day length during the year, and shift working, may influence the skin symptoms of psoriasis. The differences in the psoriasis activity scores during the year in different countries can be not only influenced by environmental factors but also by cultural activities. The role of vitamin D level in PsV patients is still questionable. New data are emerging on the possible relationship between the season of treatment initiation and treatment efficacy. Further understanding of the seasonality of psoriasis may improve healthcare resource planning in disease management.

## Figures and Tables

**Figure 1 ijms-25-11670-f001:**
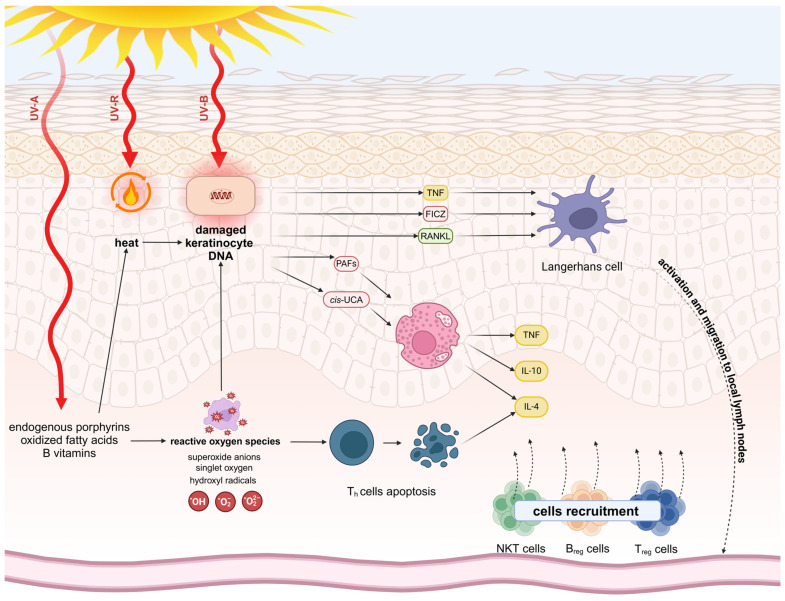
Influence of UV-R on the skin. When UV electromagnetic waves reach the skin, a small part of them are reflected, while the majority penetrate the skin and are absorbed by different molecules [77]. UV-B penetrates the skin to a depth of about 0.1 mm and UV-A to a depth of about 0.8 mm [79]. While UV-B is absorbed by most of the molecules within the skin, UV-A is not absorbed by DNA and most proteins but can be absorbed by endogenous porphyrins, oxidized fatty acids, and B vitamins [77,80]. The energy of absorbed UV is converted into heat, fluorescence, and chemical changes in the absorbing molecules, causing the formation of reactive oxygen species (ROS) such as superoxide anions, singlet oxygen, and hydroxyl radicals [77,80]. ROS can damage DNA and promote T-cell apoptosis [77,81,82,83,84]. Damaged keratinocytes produce and release several immunomodulators, such as tumor necrosis factors (TNF), 6-formylindolo[3,2-b]carbazole (FICZ), receptor activator of NF-κB ligand (RANKL), platelet-activating factors (PAFs), cis-urocanic acid (cis-UCA), and IL-10 [85,86,87]. TNFs, FICZ, and RANKL cause activation and migration of Langerhans cells to local lymph nodes [85,86,88,89]. PAFs and cis-UCA induce further release of TNFs, IL-10 and IL-4, which supports recruiting of cells such as Natural Killers (NKT) cells and regulatory B (Breg) and T (Treg) cells, which promote humoral and cell-mediated immunosuppression [85,90,91,92,93,94]. This figure was created with BioRender.com.

**Table 1 ijms-25-11670-t001:** Examples of gene polymorphisms associated with psoriasis and their clinical implications.

Described Polymorphism	References	Clinical Features/Implications
*HLA-B*27* *HLA-B*39* *HLA-B*38* *HLA-B*08*	[26,42,43,44]	Higher risk of PsA development.
*HLA-B*08*	[26,44]	Asymmetric sacroiliitis, peripheral arthritis ankylosis, and increased joint damage.
*HLA-B*27*	[26,44]	Symmetric sacroiliitis, dactylitis, and enthesitis development.
*HLA-C*06:02*	[28,45]	An earlier onset of PsV and a later onset of PsA. Photosensitive psoriasis.
*CARD14*	[19,31,32,33]	Psoriasis and/or features of PRP and GPP. Photosensitive psoriasis.
*FTO* *CALCR* *AC003006.7*	[41,46]	PsV associated with obesity.

Abbreviations: PsA—psoriatic arthritis; PsV—psoriasis vulgaris; PRP—pityriasis rubra pilaris; GPP—generalized pustular psoriasis.

**Table 2 ijms-25-11670-t002:** Spectrum of solar radiation. Table adapted from Kurz et al. [77].

Light Type	Abbreviation	Wavelength [nm]
ultraviolet radiation	UV-R	100–400
	UV-C	100–280
	UV-B	280–315
	UV-A	315–400
visible light	VL	400–760
infrared radiation	IR	760–1000

Abbreviations: UV-R—ultraviolet radiation; VL—visible light; IR—infrared radiation.

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
