# Peer review of "Psoriasis and Seasonality: Exploring the Genetic and Epigenetic Interactions"

_ijms, 2024, doi:10.3390/ijms252111670_

Round 1
Reviewer 1 Report
Comments and Suggestions for Authors
Review on Seasonality of Psoriasis: From Genes to the Clinical Phenomena by Michał Niedźwiedź et al.
This manuscript is an excellent, detailed review devoted to the influence of genetic, epigenetic, and environmental factors on the seasonality of psoriasis vulgaris.
I did not make any major work recommendations.
Minor comments:
1) Page 1, line 27 – ten to forty percent..., but on the page 2, line 57 – 30-35% - it would be great to unify the numbers.
2) P2, line 67 – a word “chromosomal” should be inserted between “fifteen” and “regions”
3) P3, line 96-97 – this sentence would benefit in the change of order of listed genes, since due to the last one AC003006.7 one might conclude that new sentence is starting by ”7 genes were found…”.
4) Table 1 – It seems to me that Table 1 could be supplemented with references to publications already mentioned in the text of the manuscript.
5) P3, line 122-124 – sentence starting with Chandra et al. … - comparing should be replaced to “compared them to”
6) P4, line 159 – replace comma from “In several, studies miRNA-146a..” to In several studies, miRNA-146a..”
7) P4+5, line 181-182 – did the author mean keratinocytes (cells of the skin) or keratocytes (cells of the cornea) ?
8) P5+9, lines 187, 191, 389, 390 – IFN should be instead of INF
9) P5, table – should be numbered as 2; moreover, it is identical together with few sentences, which can be found in the cited paper (68 - Kurz, B.; Berneburg, M.; Baumler, W.; Karrer, S. Phototherapy: Theory and practice. J Dtsch Dermatol Ges 2023, 21, 882-897, 700 doi:10.1111/ddg.15126.). Moreover, the numbers of wavelengths reaching the Earth, do not match. If radiation with wavelength over 1000nm reaches the Earth, should not be there a mention how big is the portion of radiation above 1000nm to 2500nm in whole spectra reaching the Earth?
10) P6, figure 1 – in the figure, only UV-R is mentioned but the legend uses UV-A, and -B. It should be unified. Sentences at lines 218-221, with an exception of “Singlet oxygen promotes T-cell apoptosis „ – should be deleted since they describe things not seen in the figure.
11) P6, lines 233,238 – the wavelength 311nm behind UV-B should be either in brackets or the phrase “UV-B of 311 nm” should be used.
12) P7, line 284 – The mention of the study investigating TEWL and SCH in psoriatic as well atopic dermatitis patients. But consequently, no mention about findings in AD patients. This should be added.
13) P11, figure 2 – relatively complex figure, which lacks legend, that would explain the content of the figure in similar way as in figure 1. Moreover, it should be increased since it is less readable than figure 1.
14) P13, table – again numbered as 1. In addition, there is no reference in the manuscript to a second or third table. In addition, this third table would benefit from reducing the size of the letters and simplifying the conclusions, such as seasonality -Yes/No, percentages in numbers instead of words, etc., since it is relatively large.
15) P16, line 538-540 – the sentence “New data is emerging investigating the possible connection between the season of starting the therapy and the effectiveness of treatment” should be reformulated for better clarity, like e.g. “New data are emerging on the possible relationship between season of treatment initiation and treatment efficacy. “
Author Response
Dear Reviewer,
Thank you so much for taking the time to review our paper. Your insightful comments and constructive feedback have been invaluable in improving the quality of our paper. We truly appreciate your expertise and dedication.
Here we attach the following responses to your comments:
Reviewer 1
Comments 1: Page 1, line 27 – ten to forty percent..., but on the page 2, line 57 – 30-35% - it would be great to unify the numbers.
Response 1: Thank you for pointing this out. The numbers were unified.
Comments 2: P2, line 67 – a word “chromosomal” should be inserted between “fifteen” and “regions”
Response 2: The word chromosomal was inserted as it was recommended.
Comment 3: P3, line 96-97 – this sentence would benefit in the change of order of listed genes, since due to the last one AC003006.7 one might conclude that new sentence is starting by ”7 genes were found…”.
Response 3: The AC003006.7 is a name of the gene. We rephrased the sentence to make it clearer.
Comments 4: Table 1 – It seems to me that Table 1 could be supplemented with references to publications already mentioned in the text of the manuscript.
Response 4: References were added to Table 1.
Comment 5: P3, line 122-124 – sentence starting with Chandra et al. … - comparing should be replaced to “compared them to”
Response 5: The sentence was rephrased according to the Reviewer’s comments.
Comment 6: P4, line 159 – replace comma from “In several, studies miRNA-146a..” to In several studies, miRNA-146a..”
Response 6: The sentence was rephrased according to the Reviewer’s comments.
Comment 7: P4+5, line 181-182 – did the author mean keratinocytes (cells of the skin) or keratocytes (cells of the cornea) ?
Response 7: Thank you for noticing it. We changed the word keratocytes to keratinocytes.
Comments 8: P5+9, lines 187, 191, 389, 390 – IFN should be instead of INF
Response 8: The INFs were changed to IFNs.
Comments 9: P5, table – should be numbered as 2; moreover, it is identical together with few sentences, which can be found in the cited paper (68 - Kurz, B.; Berneburg, M.; Baumler, W.; Karrer, S. Phototherapy: Theory and practice. J Dtsch Dermatol Ges 2023, 21, 882-897, 700 doi:10.1111/ddg.15126.). Moreover, the numbers of wavelengths reaching the Earth, do not match. If radiation with wavelength over 1000nm reaches the Earth, should not be there a mention how big is the portion of radiation above 1000nm to 2500nm in whole spectra reaching the Earth?
Response 9: The Table number was changed. Due to the similarity of the table we added the annotation about adaption of the table from Kurz et al., the sentences 203-207 were rephrased.
Comments 10: P6, figure 1 – in the figure, only UV-R is mentioned but the legend uses UV-A, and -B. It should be unified. Sentences at lines 218-221, with an exception of “Singlet oxygen promotes T-cell apoptosis „ – should be deleted since they describe things not seen in the figure.
Response 10: The figure and its legends were improved. The mentioned sentences were shortened according to Reviewers comments.
Comments 11: P6, lines 233,238 – the wavelength 311nm behind UV-B should be either in brackets or the phrase “UV-B of 311 nm” should be used.
Response 11: The sentences were improved according to the Reviewer’s comments.
Comments 12: P7, line 284 – The mention of the study investigating TEWL and SCH in psoriatic as well atopic dermatitis patients. But consequently, no mention about findings in AD patients. This should be added.
Response 12: A discussion of TEWL and SCH in atopic dermatitis would need to be expanded by a large section. Due to the fact that we discuss psoriasis in the article, we decided to focus only on PsV patients and healthy individuals.
Comments 13: P11, figure 2 – relatively complex figure, which lacks legend, that would explain the content of the figure in similar way as in figure 1. Moreover, it should be increased since it is less readable than figure 1.
Response 13: The Figure 2 was enlarged. The legend was expanded with a more detailed description.
Comments 14: P13, table – again numbered as 1. In addition, there is no reference in the manuscript to a second or third table. In addition, this third table would benefit from reducing the size of the letters and simplifying the conclusions, such as seasonality -Yes/No, percentages in numbers instead of words, etc., since it is relatively large.
Response 14: Number of the table was changed. The references to second and third table were added to the text. The font size within the table was decreased. The text within “Conclusion” column was shortened.
Comments 15: P16, line 538-540 – the sentence “New data is emerging investigating the possible connection between the season of starting the therapy and the effectiveness of treatment” should be reformulated for better clarity, like e.g. “New data are emerging on the possible relationship between season of treatment initiation and treatment efficacy. “
Response 15: The sentence was improved according to the Reviewer’s comments.
Once again, we would like to thank you for your time and effort in reviewing our article.
With kind regards
Authors
Reviewer 2 Report
Comments and Suggestions for Authors
This study aimed to understand the intricate interplay between seasonality, environmental factors, and the genetic and epigenetic landscape of psoriasis. This manuscript targeted interventions and personalized management strategies for individuals living with psoriasis.
1. This manuscript is interest and many information are present. This manuscript discusses many factors related to the occurrence of psoriasis, not just seasonality. Therefore, the title of this manuscript should be modified to be more relevant to the content of the manuscript.
2. A review manuscript must apply new information and research trend of the field to the readers. Most of the reference cite in this manuscript were published before 2020. The reference and information present in this manuscript must renew and update.
3. The factors related to the occurrence of psoriasis discussed in the article should be linked to and discussed with seasonal factors.
4. Line 170-171. “Dopico et al. [55] found that 23% of the genome (5.136 unique genes out of 22.822 170 genes tested) show significant seasonal differences in expression.” The number in this sentence is correct? Please recheck.
5. In Table 1 (must revise to Table 2), the definition of “non data” and ND must explain in the note of the Table.
6. The abbreviation must define at the first time shown in the text.
Comments on the Quality of English LanguageMinor editing of English language required.
Author Response
Dear Reviewer,
Thank you so much for taking the time to review our paper. Your insightful comments and constructive feedback have been invaluable in improving the quality of our paper. We truly appreciate your expertise and dedication.
Here we attach the following responses to your comments:
Comment 1: This manuscript is interest and many information are present. This manuscript discusses many factors related to the occurrence of psoriasis, not just seasonality. Therefore, the title of this manuscript should be modified to be more relevant to the content of the manuscript.
Response 1: According to your recommendations we propose new title for the paper: “Psoriasis and Seasonality: Exploring the Genetic and Epigenetic Interactions”
Comments 2: A review manuscript must apply new information and research trend of the field to the readers. Most of the reference cite in this manuscript were published before 2020. The reference and information present in this manuscript must renew and update.
Response 2: According to the Reviewer’s suggestion, we have added new references from the last 3 years.
Comments 3: The factors related to the occurrence of psoriasis discussed in the article should be linked to and discussed with seasonal factors.
Response 3: The fragment discussing environmental factors are linked within the “Seasonality of psoriasis” section. Additionally, those links are discussed in the description of Figure 3.
Comments 4: Line 170-171. “Dopico et al. [55] found that 23% of the genome (5.136 unique genes out of 22.822 170 genes tested) show significant seasonal differences in expression.” The number in this sentence is correct? Please recheck.
Response 4: The numbers were corrected.
Comments 5: In Table 1 (must revise to Table 2), the definition of “non data” and ND must explain in the note of the Table.
Response 5: Abbreviations were added to the note of the tables.
Comments 6: The abbreviation must define at the first time shown in the text.
Response 6: Abbreviations were defined within text.
The English language was improved by a native speaker.
Once again, we would like to thank you for your time and effort in reviewing our article.
With kind regards
Authors